# LEARNING OBJECTS FROM PIXELS

## ABSTRACT

We show how discrete objects can be learnt in an unsupervised fashion from pixels, and how to perform reinforcement learning using this object representation.

More precisely, we construct a differentiable mapping from an image to a discrete tabular list of objects, where each object consists of a differentiable position, feature vector, and scalar *presence* value that allows the representation to be learnt using an attention mechanism.

Applying this mapping to Atari games, together with an *interaction net*-style architecture for calculating quantities from objects, we construct agents that can play Atari games using objects learnt in an unsupervised fashion. During training, many natural objects emerge, such as the ball and paddles in *Pong*, and the submarine and fish in *Seaquest*.

This gives the first reinforcement learning agent for Atari with an interpretable object representation, and opens the avenue for agents that can conduct object-based exploration and generalization.

## 1 INTRODUCTION

Humans are able to parse the world as a collection of objects, that are discrete, persistent, and can be interacted with. Humans can use this representation for planning, reasoning, and exploration. When playing a game such as *Montezuma's Revenge* in Atari, a human can identify the different objects, such as an avatar that moves in a 2-D plane, a rolling skull, and a key. Even if they do not know initially what to do, they can explore the state space using the prior knowledge that objects persist, move around contiguously, and can interact with other objects in local proximity.

This explicit representation of objects and prior knowledge is missing from artificial reinforcement learning agents, such as DQN (Mnih et al. (2015)). Although architectures such as DQN attain superhuman performance on many games, in particular those whose reward signal is dense (see e.g., Bellemare et al. (2016)), its performance on games with sparse rewards, or greater planning complexity, is often below that of humans. Perhaps explicit object knowledge is one missing ingredient, which would allow for more powerful exploration than existing epsilon-greedy methods (that simply execute a random walk in action space).

In this paper we set forth a method to learn objects from pixels in an unsupervised manner. By an object representation, we mean a "tabular" representation, where there is a list of objects, and each object has a position and a set of features (represented by a vector).

Learning such a representation from input pixels is a non-trivial challenge. The space of possible inputs is a connected manifold, but the space of object representations is disconnected; for example, there is no continuous transformation from 4 objects to 5. We address this challenge by introducing an object *presence* value between 0 and 1, which is a continuous relaxation of whether an object is present or not.

We give a method of tracking the same object across multiple frames (object persistence), and give an architecture that can perform calculations using the object representation. We test this model in the Atari domain, and show that it is possible to do reinforcement learning on a learnt object representation. Objects such as the ball and paddles in *Pong*, and the submarine and fish in *Seaquest*, emerge naturally without supervision. We give results and insights into how best to calculate global values from a collection of objects using an "interaction net" style architecture, where calculations are invariant to object order.

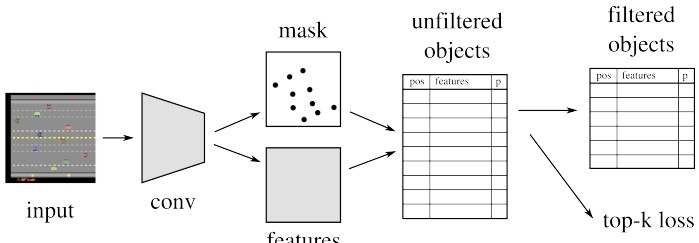

Figure 1: The pixel-to-object model.

## 2 RELATED WORK

The main contributions of our work are a method for learning an object representation from pixels without supervision of what constitutes an object, and how to use this representation for doing reinforcement learning. There are various papers that have looked at extracting objects from scenes without object labels, or using an object representation to perform calculations.

The model in *Attend, Infer, Repeat: Fast Scene Understanding with Generative Models* (Eslami et al. (2016)) produces objects from pixels in an unsupervised manner. For example, on a dataset of images consisting of a few scattered MNIST digits, their model learns to represent each digit as an object. Their model is based on recurrence, at each step giving an object location, and as such is able to deal with overlapping objects (which our model cannot). However, our model can handle larger images with more objects, possibly because we avoid both recurrence and the use of spatial transformers, which may have issue "homing in" on undiscovered objects: we can detect tens of objects in Atari images, whereas the datasets they use only have up to three objects. Another (completely different) approach to detecting objects is given by Stewart and Ermon (2017), using the prior that any detected object must obey physical laws, such as following a parabola when in free fall. This relies on having access to a contiguous sequence of frames.

There are various models that have a "two-stage" architecture. Garnelo et al. (2016) propose a class of architectures for doing reinforcement learning, consisting of a "neural back end" and a "symbolic front end", where the neural back end is responsible for recognizing objects from low-level pictures, and the symbolic front end does fast generalizable reinforcement learning using these objects. More specific to our model, there are various models that use an interaction network architecture on top of a convolutional stack from an input image (as we do). *Visual interaction networks* (Watters et al. (2017)) learns to extract a few objects from pixels and predict future state (such as bouncing balls) using a *supervised* training loss. *A simple neural network module for relational reasoning* (Santoro et al. (2017)) learns to answer questions from pixels that involve spatial reasoning; the intermediary representation fed into the interaction network is not quite a set of objects, but rather the output from the final convolutional layer.

## 3 THE MODEL

Here we describe the differentiable mapping from pixels to a list of objects, which produces a list

$$(x_i, f_i, p_i)_{i=1}^k,$$

where for the $i$th object, $x_i \in \mathbb{R}^2$ is its position, $f_i \in \mathbb{R}^{n_f}$ is its features (for some $n_f \geq 1$), and $p_i \in [0, 1]$ is its "presence", which should be interpreted as a continuous relaxation of whether the $i$th object is present or not. The object representation acts as a *bottleneck*; objects are then learnt by minimizing a loss function using the object representation.

The model is depicted in Figure 1. Here we give a high-level overview, and below we are more precise. To obtain the list of objects, the input pixels go through a series of transformations to obtain the final list. First, the pixels are mapped through a convolutional input network, to obtain two tensors: a 2-D *mask* tensor, and a 3-D *features* tensor, with the same height and width. The mask tensor is the additional output of a sigmoid layer, so that it is scaled between 0 and 1. Second, this mask tensor is used to identify object positions: for each local maxima in the mask, we sample

the object position from a diagonal Gaussian whose statistics match the mask in a small window centered at the local maxima. This gives a list of object positions, and we associate object features with these by convolving the mask with the feature tensor, and use the value of the local maxima for the presence value of each object. Third, we take the top $k$ of these objects by presence value for computational efficiency further down the "pipeline". Fourth, we add noise to the presence values (before they are used by downstream loss functions); this helps in exploration and learning sharp mask values. Finally, we include an optional loss term that to discourage useless objects from appearing.

We now describe the model in mor edetail and more formally. First, map the input pixels through a convolutional stack, with a final sigmoid activation in the first case, to produce a pair of tensors:

- A *mask* tensor $m \in [0, 1]^{h \times w}$.
- A *features* tensor $f \in \mathbb{R}^{h \times w \times n_f}$.

Roughly speaking, the local maxima of the mask give the approximate object positions, and the values of the mask at these points will give the presence values. However, the local maxima are integer coordinates and not differentiable, so we modify this by taking the object position as the weighted average in a small window around the local maxima, as follows.

For coordinates $(a, b)$ and $r \geq 1$, let $B_r(a, b) = \{(c, d) : |a - c|, |b - d| \leq r\}$ be the window (or $\ell_\infty$–ball) around $(a, b)$ with radius $r$. (In our application to Atari, we take $r = 2$.) With the notation $[h] = \{1, ..., h\}$, let

$$M = \{(a, b) \in [h] \times [w] : m_{a,b} \geq m_{c,d} \text{ for all } (c, d) \in B_r(a, b)\}$$

be the set of local maxima. (In our implementation, we add a small uniform random number to each pixel of the mask to break ties, ensuring that there is at most one maxima in each window.) For each $(a, b) \in M$, define the mean position $x_\mu(a, b) \in \mathbb{R}^2$ as the center of mass within this window:

$$x_\mu(a, b) = \Big( \sum_{(c,d) \in B_r(a,b)} m_{c,d} \Big)^{-1} \sum_{(c,d) \in B_r(a,b)} m_{c,d} \cdot [c, d]$$

We could use $x_\mu(a, b)$ as an object position. However, it is better to sample the object position from a diagonal Gaussian with statistics of the mask inside the local window, because this encourages the mask to produce sharp values about the actual object positions, and encourages exploration in the latent space of positions. Thus define $x_\sigma(a, b)^2 \in \mathbb{R}^2$ to be the diagonal of the covariance matrix of the mask within the window $B_r(a, b)$. In sampling the position, we subtract a small constant from this, so it is possible to have fractional (non-integer) positions with zero standard deviation[1]. We need to subtract at least 0.5, and empirically 1 works well. Thus, for local maxima $(a, b) \in M$, we take the final position to be the Gaussian sample

$$x(a, b) \sim \mathcal{N}\big(x_\mu(a, b), (\max(x_\sigma(a, b) - 1, 0))^2\big). \tag{1}$$

We can make this a differentiable function of $x_\mu$ and $x_\sigma$ by the reparameterization trick (introduced in the context of variational inference independently by Kingma and Welling (2014), Rezende et al. (2014), and Titsias and Lazaro-Gredilla (2014)).

The object feature vector is the average of the feature tensor in the window about the maxima, weighted by the mask,

$$f(a, b) = \Big( \sum_{(c,d) \in B_r(a,b)} m_{c,d} \Big)^{-1} \sum_{(c,d) \in B_r(a,b)} m_{c,d} f_{c,d}. \tag{2}$$

Putting this together, each $(a, b) \in M$ indexes an object, with position $x(a, b)$ given by (1), feature vector $f(a, b)$ given by (2), and presence $q(a, b)$ equal to $m(a, b)$.

Select the top $k$ values of $m(a, b)$ with $(a, b) \in M$ (see motivation below) to produce a list of objects with pre-modified presence values

$$(x_i, f_i, q_i)_{i=1}^k.$$

---

[1]For example in one dimension, the position 0.5 could be represented by the spatial average of $m_0 = m_1 = 1$; however, this has standard deviation 0.5; we need to sample from a tighter distribution in order to get a deterministic fractional position.

(If there are less than $k$ objects, then pad using arbitrary values for the position and features, and with presence values equal to zero.)

To produce the final presence values $p_i$, we use the concrete transformation (Maddison et al. (2017) and Jang et al. (2017)) with temperature equal to 1 (see justification below), which in this case simplifies to

$$p_i = \frac{q_i \log u_1}{q_i \log u_1 + (1 - q_i) \log u_0},$$
(3)

where $u_0$ and $u_1$ are independent uniform random variables in the range $[0, 1]$ (independently for each object).

## 3.1 USING THE PRESENCE VALUE

We require any function defined on the objects to be smoothly gated on the presence (so that an object with presence 0 has no effect on the output). In this way, minimizing a loss defined on this function encourages useful objects to have larger presence (i.e., to emerge). See Sections 4 and 5 for examples of this gating.

## 3.2 WHY DO WE ADD "NOISE" TO THE PRESENCE VALUES?

In reality, objects are either "fully present" or not present at all; it does not make sense to have an object half present. Thus we wish the presence values to go to 0 or 1. To achieve this, we add noise to the presence values from the mask before they are used to gate any functions. The concrete transformation used satisfies the property that presence values close to 0 or 1 tend not to change much, but presence values near 0.5 can change a lot. In this way, if an object is useful, then learning encourages the presence value to go to 1 where the output is a more deterministic function of the object. The concrete transformation as described works well (much better than with no noise at all), and experimentally the best temperature to take is 1. We did not investigate annealing.

## 3.3 WHY DO WE TAKE THE TOP K?

There are a variable and potentially large number of objects indexed by the local maxima of the mask. Objects with small presence value do not have much effect on the output, and it is computationally wasteful to retain them. Furthermore for our TensorFlow implementation, it is convenient to have a fixed number of objects. So we simply take the top $k$ objects for a sufficiently large $k$ (we take $k = 16$). This is not the same as assuming there are exactly $k$ objects, since having a presence close to zero is the same as not having an object.

Taking a fixed number objects introduces an obvious risk: objects that are useful to minimize the loss, but have an initially small presence value, may never be "seen", as there is nothing explicitly discouraging useless objects from occupying the top $k$ slots[2]. Other approaches for sampling $k$ objects cannot ameliorate this issue either: any reasonable sampling method will not consistently select objects for which there are at least $k$ other objects with larger presence.

Thus to discourage useless objects from appearing, it is sometimes necessary to add a loss penalty to discourage the "expected slot occupancy" going beyond some amount $\alpha$,

$$\ell_{\text{top}} = \max \left( 0, \sum_{i=1}^{k} p_i / k - \alpha \right).$$
(4)

(In our application to Atari, we take $\alpha = 0.5$.)

We have experimented with other schemes, such as sampling objects in proportion to their presence value, but have found that these more complex schemes do not perform as well.

---

[2]We found that useless objects tend not to be learnt early on in training. We suspect that if objects are not useful for reducing the loss, then they are typically harmful, since the loss function has not yet learnt to ignore them. However after further training, the loss function can learn to ignore them, and useless objects can reappear.

### 3.4 COMPARISON WITH OTHER ATTENTION MODELS

It is interesting to compare the attention with loss scheme used here with that in Shazeer et al. (2017) (used for selecting network modules to perform forward calculations), as it highlights what is and is not important for us. Probably the main difference is that in our use case, it is far more important that *all* the useful objects are attended to (for example, in playing *Pong*, we always want to aware of the ball); after this, remaining slots can be filled with only potentially useful objects. Whereas in their model, if a quite useful network module is not selected, then many other modules will still perform well, and indeed they do not always want to select the currently most useful modules to avoid having only a few modules becoming "experts". These differences mean that we always have to select the top $k$ objects (before noise has been added), and otherwise add a small loss term to discourage useless objects from appearing; there is no "load balancing loss".

### 3.5 MATCHING OBJECTS IN SUCCESSIVE FRAMES

For some applications, it is useful to identify the same object across successive frames. We do this by constructing a $2k \times 2k$ bipartite graph with vertex classes $A = \{a_1, \ldots, a_{2k}\}$ and $B = \{b_1, \ldots, b_{2k}\}$ with edge weights defined below, and find a *minimal* edge weight perfect matching. Vertices with index $\leq k$ correspond to objects, and those with index $\geq k + 1$ correspond to non-objects. For example, suppose $a_i$ and $b_j$ are matched; if $1 \leq i, j \leq k$, then $i$ and $j$ correspond to the same object; if on the other hand $1 \leq i \leq k < j$, then object $i$ is not present in the second frame. The perfect matching can be found in polynomial time (e.g., by the Hungarian algorithm).

The following constants control the matching thresholds.

$$s = \min\{\text{height, width}\} \qquad\qquad \text{length scale}$$
$$c_t = 0.02 \qquad\qquad \text{matching threshold}$$
$$c_p = 0.03 \qquad\qquad \text{presence weight coefficient}$$

The weights of the bipartite graph are

$$w_{i,j} = \begin{cases} ||(x_i - x'_j)/s||_2^2 + c_p|\log(p_i/p'_j)| & \text{for } 1 \leq i, j \leq k, \\ 0 & \text{for } k + 1 \leq i, j \leq 2k, \\ c_t & \text{otherwise.} \end{cases} \qquad (5)$$

Observe that the constants have been judiciously chosen so that, for example, object $i$ in the first frame will never be matched to object $j$ in the second frame if

$$||x_i - x'_j||_2 > s\sqrt{2c_t} = 0.2\min\{\text{height, width}\}, \text{ or}$$
$$p'_j < p_i \exp\{-2c_t/c_p\} \approx p_i/4.$$

A minimal weight matching also has a probabilistic interpretation. Ignoring the presence term in (5) and conditioning on there being no unmatched objects, then such a matching can be interpreted as the maximal likelihood assignment of objects, if the objects in the second frame are sampled from Gaussian random variables with identical standard deviation and means equal to the positions of the objects in the first frame (under some permutation of the object order).

We have not included features in the weights in (5), since it is not clear what scaling to give them, although adding such a term may further improve the quality of the matching.

This matching mechanism is entirely deterministic with no learnable parameters, and we have found it works well in practice. However, we could give a learnable bias to the edge weights, and interpret the edge weights as log probabilities and use ideas from the concrete distribution to allow the network to learn how to match objects.

For an example of object matching in successive frames in this model, see the video at `https://goo.gl/AEXEQP` (matched objects get the same colour in successive frames).

## 4 CALCULATIONS USING THE OBJECT REPRESENTATION

In this section we describe how to compute quantities of interest (such as Q-values as in DQN, Mnih et al. (2015) for reinforcement learning) using the object representation, satisfying:

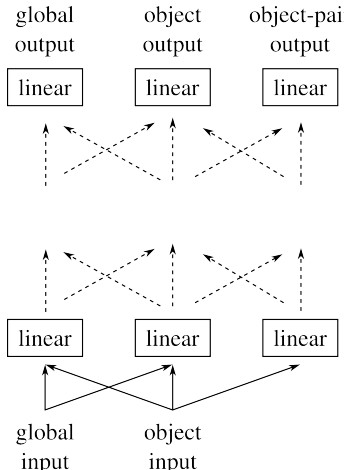

Figure 2: Our calculation architecture consists of three streams: a global stream, a per-object stream, and a per-object-pair stream. Each "layer" consists of mixing, followed by a linear transformation and a ReLU activation (for non-final layers). The layers are repeated several times.

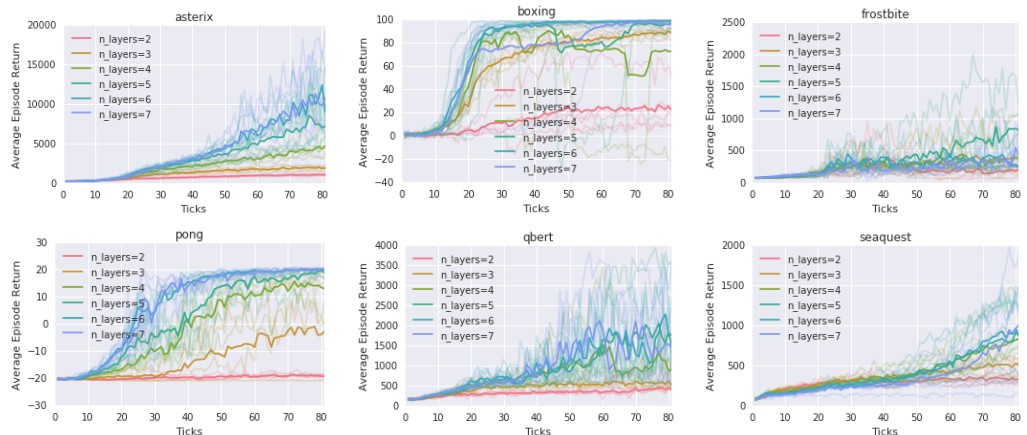

Figure 3: One surprising lesson was that the interaction network-style architectures need to be quite deep to be effective (and learn Q-values), despite having "clear" representations for input. This is surprising as DQN and A3C have success with relative shallow architectures (4 to 5 layers including both convolutional and fully connected layers). These graphs show the effect on average episode score on some of the Atari games, according to the number of layers in the interaction network alone. The x-axis scale is one tick is 800k agent steps.

1. The output should be invariant to object order.

2. The output should be invariant to objects with zero presence, and otherwise smoothly gated on the presence.

3. The architecture should support recurrence.

(Recurrence is required for reinforcement learning. For example, it is is important to know which direction the ball is travelling in in *Pong*. DQN avoids recurrence by stacking together input frames, however, this makes it much harder for the architecture to learn objects, which no longer have a single well-defined location.)

We use ideas introduced in interaction networks (Battaglia et al. (2016)), also introduced in parallel by Chang et al. (2016), for the architecture. Precise details are given in Appendix B.3.

Figure 4: `https://goo.gl/AEXEQP`. Learning to play *Seaquest* from objects (from left to right, images are: inputs, decoded, mask, objects, and Q-values). The "oxygen remaining" bar is not learnt in this run, but the submarine and other objects are learnt.

The architecture is composed of identical layers (with separate parameters), that operate on three streams of information. The input and output of each layer is a triple $(A, B, C)$, where $A \in \mathbb{R}^a$, $B \in \mathbb{R}^{k \times b}$, and $C \in \mathbb{R}^{k \times k \times c}$; these represent global quantities, per-object quantities, and per-object-pair quantities respectively (see Figure 2). Typically we have $a \gg b \gg c$ for computational expediency (we take $a = 512, b = 256, c = 64$ for reinforcement learning in Atari), although the input to the first layer has $a = c = 0$ and $b = n_f + 2$ being the concatenation of the object features and its position, and the output of the final layer has $a$ equal to the size of quantity required for the global output (e.g., the number of actions for Q-learning).

Each layer performs its calculation by mixing its three inputs (broadcasting, reducing, and concatenating as appropriate), before applying a learnable linear transformation. In the linear transformation, we achieve invariance to object order by treating the object dimension(s) as batch dimensions. We use ReLUs for internal activations. The reduction operation is an average over an object axis as in interaction networks, except we also gate on the object presence so the output is invariant to objects with presence 0. We also found it useful to allow self-gating of the network, as this allows the network to ignore objects that are not useful for calculating Q-values, without penalizing them if they are useful in reconstructing the input as described in Section 5.

To make the network recurrent, we maintain a per-object state vector (of size 20 for Atari), which is updated using LSTM-style gates: if the $i$th object has state $s_i^{t-1}$ at time $t - 1$, then its state in the next frame is

$$s_i^t = \sigma(f_i^t) s_i^{t-1} + \sigma(i_i^t) \tanh(j_i^t), \tag{6}$$

where $f_i^t$, $i_i^t$, $j_i^t$ are output from the per-object stream, and this state is fed back into the per-object stream at the next time step. (Objects are paired in adjacent frames using the pairing described in Section 3.5, and objects that appear in a given frame have state initialized to zero.)

We chose this architecture because it is fairly homogeneous (there are two axes of freedom: the number of layers, and the sizes of the streams), and it allows the input and output of global and per-object information, such as object state in recurrence, global Q-values, or per-object Q-values for applications such as object-based exploration.

## 5 PIXEL DECODER

An easy and useful way of learning objects from pixels is to use them to reconstruct the input. We do this by drawing each object using a stack of transpose convolutional layers (Zeiler et al. (2010)) with the object features as inputs, and composing these with a learnt static background, positioned using the object position and gated using the presence values. The decoder loss $\ell_{\text{decoder}}$ is the $l_2$ pixel reconstruction loss. This model works well for many of the Atari games. Implementation details are in Appendix B.2.

## 6 APPLICATION: REINFORCEMENT LEARNING

As an example application, we show here that it is possible to combine these models to build a reinforcement learning agent. We use distributional Q-learning as in Bellemare et al. (2017) for speed of learning, with outputs calculated from the object representation using the architecture described in Section 4. For Atari games, it is beneficial to use the object reconstruction loss $\ell_{\text{decoder}}$ described in

Section 5 to bootstrap object identification. We also include the top-$k$ loss $\ell_{\text{top}}$ defined in Section 3.3, which is needed in some environments to prevent phantom (useless) objects from appearing. Thus we train our agents using the combined loss

$$\ell = \ell_{\text{rl}} + c_{\text{decoder}} \cdot \ell_{\text{decoder}} + c_{\text{top}} \cdot \ell_{\text{top}} \tag{7}$$

for appropriate constants $c_{\text{decoder}}$ and $c_{\text{top}}$. Examples of learnt objects are presented in Appendix A.1, and full details of the RL implementation are in Appendix B.4.

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

## A    RESULTS

### A.1    EXAMPLES OF OBJECTS LEARNT

These images (best viewed in colour) show examples of objects learnt for some of the more commonly reported Atari games. On the whole, the Atari games shown below are quite amenable to learning objects; games with moving backgrounds struggle a lot more to learn objects.

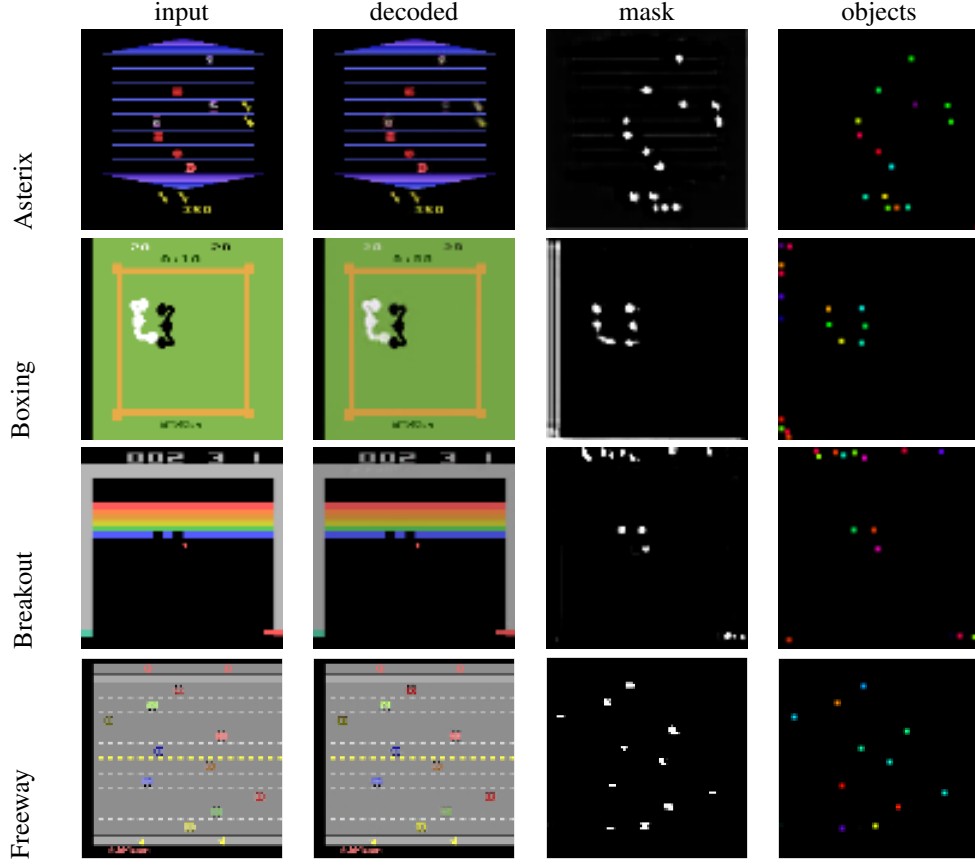

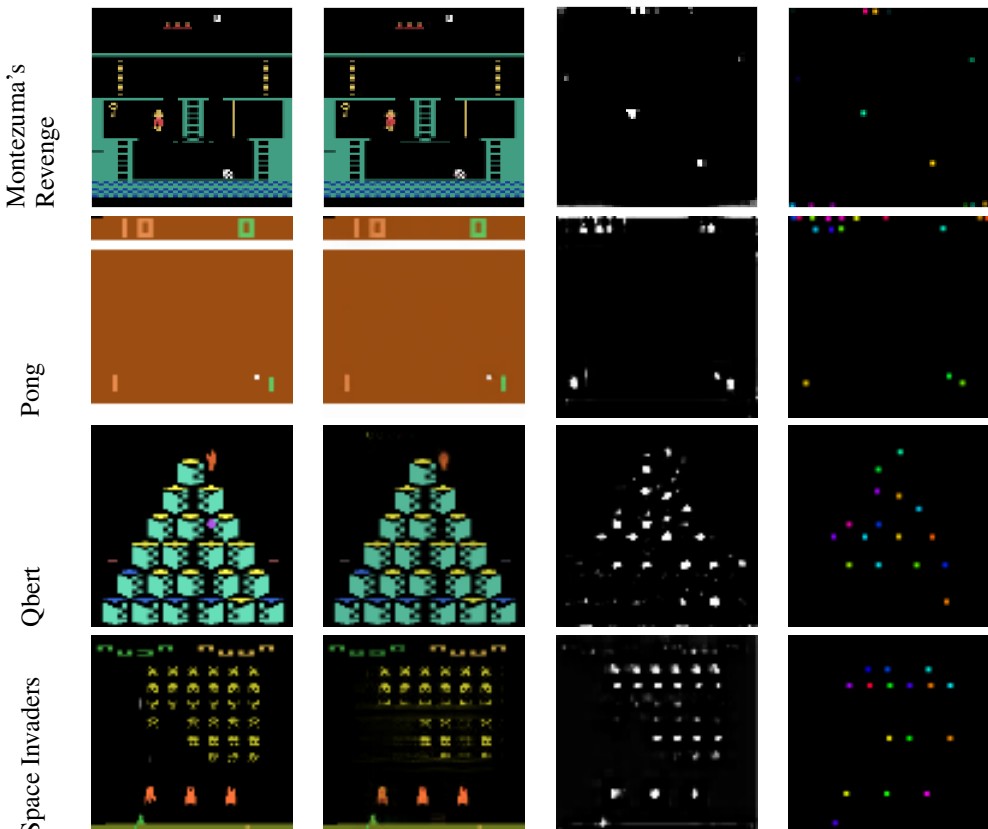

# B IMPLEMENTATION DETAILS

Here we describe the architecture used for Atari. Following DQN (Mnih et al. (2015)), we downscale the input image to $84 \times 84$, but we retain colour and do not stack frames. Thus in what follows, the input is an $84 \times 84 \times 3$ tensor $I$. The per-object feature vector has size $n_f = 4$.

## B.1 ENCODER ARCHITECTURE

The encoder is the most expensive part of the model, as the output mask has the same size as the input, with large receptive field. (Another option is to use a mask with a smaller resolution than $84 \times 84$, however, this would also reduce the power to distinguish objects close together.)

For a tensor $X$, define $\mathrm{Conv}(X, n, t, s)$ to be the output of a padded convolution on $X$ with $n$ features, kernel size $t \times t$ and stride $s$, and define $\mathrm{scale}(X)$ to be the scaling of $X$ so that its height and width are both $84$. Let ReLU be the rectifier nonlinearity. Let $X_1 = \mathrm{ReLU}(\mathrm{Conv}(I, 30, 5, 2)) \in \mathbb{R}^{42 \times 42 \times 30}$, $X_2 = \mathrm{ReLU}(\mathrm{Conv}(X_1, 20, 3, 2)) \in \mathbb{R}^{21 \times 21 \times 20}$, and $X_3 = \mathrm{Conv}(X_2, n_f, 3, 1) \in \mathbb{R}^{21 \times 21 \times n_f}$. The output features are then $f = \mathrm{scale}(X_3)$. Let $X_4 \in \mathbb{R}^{84 \times 84 \times (53 + n_f)}$ be the stacking of $I, X_1, X_2, X_3$ after scaling to $84 \times 84$. Use a $1 \times 1$ convolution to get $X_5 = \mathrm{ReLU}(\mathrm{Conv}(X_4, 3, 1, 1))$ with reduced number of channels. Then the output mask is $m = \sigma(\mathrm{Conv}(X_5, 1, 3, 1))$.

## B.2 DECODER ARCHITECTURE

We draw each object using a stack of transpose convolutional layers (Long et al. (2015)) with the feature vector (as a $1 \times 1 \times n_f$ tensor) as input. We use 3 layers, all with kernel size $3 \times 3$ and stride 2, and layer sizes 30, 20, $2d$, where $d = 3$ is the number of output channels. Thus each object drawing

has size $15 \times 15 \times 2d$. For each object, we split the output into two tensors $X, Y \in \mathbb{R}^{15 \times 15 \times d}$, set $X' = p\sigma(X - 4)$ and $Y' = pY$ (where $p$ is the corresponding object's presence), recombine $X'$ and $Y'$, and add together into a "buffer" with dimensions $h \times w \times 2d$ (the buffer is zero where there are no objects drawn). We include the "$-4$" term to aid learning by biasing the objects towards not making the background too "transparent" early on (see below). When adding an object into the buffer, its center is positioned at the object position, using bilinear interpolation for fractional positions.

To produce the final decoded output $D$, split the buffer along the third axis into two tensors $X, Y \in \mathbb{R}^{84 \times 84 \times d}$, and learn an additional static background $B \in \mathbb{R}^{84 \times 84 \times d}$, to give the decoded output as $D = \text{clip}(1 - X, 0, 1) \cdot B + Y$, where "clip" bounds $1 - X$ between 0 and 1. The loss is then

$$\ell_{\text{decoder}} = ||I - D||_2^2. \tag{8}$$

### B.3 CALCULATION ARCHITECTURE

The interaction network style architecture described in Section 4 consists of a repeated parameterizable layer module with non-shared parameters, operating on three streams of information: global, per-object, and per-object-pair. We describe precisely the layer architecture here.

Recall that $k$ is the number of objects. Let $\alpha, \beta, \gamma$ be the input stream sizes, and let $a, b, c$ be the output stream sizes (for all except the first and last layer, $\alpha = a$, $\beta = b$, and $\gamma = c$). The layer will be a mapping

$$f : \mathbb{R}^{\alpha} \times \mathbb{R}^{k \times \beta} \times \mathbb{R}^{k \times k \times \gamma} \to \mathbb{R}^{a} \times \mathbb{R}^{k \times b} \times \mathbb{R}^{k \times k \times c}$$
$$(A, B, C) \mapsto (\text{linear}(A', \theta_a), \text{linear}(B', \theta_b), \text{linear}(C', \theta_c)),$$

where $A', B', C'$ are functions of $A, B, C$ defined below, "linear" is an affine transformation, and $\theta_a, \theta_b, \theta_c$ are the learnable parameters of the linear transformation.

Let $(p_i)_{i=1}^k$ be the presence values. Define "reduce" as

$$\text{reduce}(B)_u = (1/k) \sum_i p_i \sigma(B_{i,1}) B_{i,u} \in \mathbb{R}^{\beta}, \tag{9}$$

$$\text{reduce}(C)_{i,u} = (1/k) \sum_j p_j \sigma(C_{i,j,1}) C_{i,j,u} \in \mathbb{R}^{k \times \gamma}, \tag{10}$$

i.e., the last object axis is averaged over (as in interaction networks), with additional gating on presence (as required), and self-gating (which improves performance).

Additionally define a restricted stream tensor for $A$ and $B$ for computational expediency: let $\tilde{A} \in \mathbb{R}^{\min(\alpha, b)}$ and $\tilde{B} \in \mathbb{R}^{k \times \min(\beta, c/2)}$ be the restrictions of $A$ and $B$ to the first $b$ and $c/2$ indices in the non-object axis respectively.

As in interaction networks, define $\text{marshall}(\tilde{B}) \in \mathbb{R}^{k \times k \times \min(2\beta, c)}$ by

$$\text{marshall}(\tilde{B})_{s,t,u} = \begin{cases} \tilde{B}_{s,u} & \text{if } u \le \min(\beta, c/2), \\ \tilde{B}_{t,u-k} & \text{otherwise}. \end{cases}$$

Further define $\text{broadcast}(\tilde{A}) \in \mathbb{R}^{k, \min(\alpha, b)}$ to be $A$ repeated along an additional first axis.

Now (finally) let $A' = \text{concat}(A, \text{reduce}(C))$, $B' = \text{concat}(\text{broadcast}(\tilde{A}), B, \text{reduce}(C))$, and $C' = \text{concat}(\text{marshall}(\tilde{B}), C)$, where concat is defined as the concatenation of the tensors over the non-object axis. This gives $f$ as defined above.

The function $f$ defines one layer of our interaction-style network. We stack these 6 times using non-shared parameters, and rectified linear unit activations in-between.

### B.4 RL SETUP

For reinforcement learning, we found it useful to train asynchronously as in the 1-step Q-learning model in Mnih et al. (2016), further modified (to boost learning) to predict distributions over Q-values as in Bellemare et al. (2017). We use 32 workers, each doing rollouts of length 12, with a

separate target network whose update period is 1000 agent steps, and whose loss is the sum along the rollouts of

$$\ell = \ell_{\mathrm{rl}} + 100\ell_{\mathrm{decoder}} + 10\ell_{\mathrm{top}}, \tag{11}$$

where $\ell_{\mathrm{decoder}}$ is defined in equation (8) below, $\ell_{\mathrm{top}}$ is defined in equation (4), and $\ell_{\mathrm{rl}}$ is the distributional RL loss defined in Bellemare et al. (2017). We use the Adam optimizer (Kingma and Ba (2014)) with a learning rate of $4 \times 10^{-5}$ and $\epsilon = 5 \times 10^{-6}$.

