# OpenReview forum: "Learning objects from pixels"
_ICLR.cc/2018/Conference — Reject_

### Official Review · AnonReviewer1 · 2017-11-24
**This paper learns to construct masks and feature representations from an input image, in order to represent objects. This is applied to the relatively simple domain of Atari games video input (compared to natural images). The paper is inadequate in that it fails to acknowledge and compare with previous methods for doing these tasks.**

**Rating:** 3
**Confidence:** 4

**Review:**

This paper learns to construct masks and feature representations from an input image, in order to represent objects. This is applied to the relatively simple domain of Atari games video input (compared to natural images). The paper is completely inadequate in respect to related work; it re-invents known techniques like non-maximum suppression and matching for tracking; fails to learn convincing objects according to visual inspection; and fails to compare with earlier methods for these tasks. (The comment above about re-invention is the most charitable intepretation -- the worst case would be using these ideas without citation.)


1) The related work section is outrageous, containing no references before 2016.  Do the authors think researchers never tried to do this task before then? This is the bad side of the recent deep nets hype, and ICLR is particularly susceptible to this. Examples include

@article{wang-adelson-94,
  author        = "Wang,  J. Y. A. and Adelson, E. H.",
  title         = {{Representing Moving Images with Layers}},
  journal       = {{IEEE Transactions on Image Processing}},
  year          = "1994",
  volume        = "3(5)",
  pages         = {625-638}
}
see http://persci.mit.edu/pub_pdfs/wang_tr279.pdf

and

@article{frey-jojic-03,
   author    = {Frey, B. J. and Jojic, N.},
   title     = {{Transformation Invariant Clustering Using the EM Algorithm}},
   journal   = {IEEE Trans Pattern Analysis and Machine Intelligence},
   year      = {2003},
   volume    = {25(1)},
   pages     = {1-17}
}
where mask and appearances for each object of interest are learned. There is a literature which follows on from the F&J paper.  The methods used in Frey & Jojic are different from what is proposed in the paper, but there needs to be comparisons.

The AIR paper also contains references to relevant previous work.

2) p 3 center -- this seems to be reinventing non-maximum suppression

3) p 4 eq 3 and sec 3.2 -- please justify *why* it makes sense to use
the concrete transform.  Can you explain better (e.g. in the supp mat)
the effect of this for different values of q_i?

4) Sec 3.5 Matching objects in successive frames using the Hungarian
algorithm is also well known, e.g. it is in the matlab function
assignDetectionsToTracks .

5) Overall: in this paper the authors come up with a method for learning objects from Atari games video input. This is a greatly restricted setting compared to real images. The objects learned as shown in Appendix A are quite unconvincing, e.g. on p 9. For example for Boxing why are the black and white objects broken up into 3 pieces, and why do they appear coloured in col 4?

Also the paper lacks comparisons to other methods (including ones from before 2016) which have tackled this problem.

It may be that the methods in this paper can outperform previous ones -- that would be interesting, but it would need a lot of work to address the issues raised above.

Text corrections:

p 2 "we are more precise" -> "we give more details"

p 3 and p 2 -- local maximum (not maxima) for a single maximum.  [occurs many times]

---

### Official Review · AnonReviewer3 · 2017-11-27
**A method for learning object representations from pixels for doing reinforcement learning. Very preliminary work. Previous work on computer vision is completely ignored**

**Rating:** 4
**Confidence:** 3

**Review:**

The paper proposes a method  for learning object representations from pixels and then use such representations for doing reinforcement learning.  This method is based on convnets that map raw pixels to a mask and feature map. The mask contains information about the presence/absence of objects in different pixel locations and the feature map contains information about object appearance.

I believe that the current method can only learn and track simple objects in a constant background, a problem which is  well-solved in computer vision. Specifically, a simple method such as "background subtraction" can easily infer the mask (the outlying pixels which correspond to moving objects)  while simple tracking methods (see a huge literature over decades on computer vision) can allow to track these objects across frames.  The authors completely ignore all this previous work and their "related work" section  starts citing papers from 2016 and onwards!  Is it any benefit of learning objects with the current (very expensive) method compared to simple methods such as  "background subtraction"?

Furthermore, the paper is very badly written since it keeps postponing the actual explanations to later sections (while these  sections eventually refer to the appendices).  This makes reading the paper very hard. For example, during the early sections you  keep referring to a loss function which will allow for learning the objects, but you never really give the form of this loss (which you should as soon as  you mentioning it) and the reader needs to search into the appendices to find out what is happening.

Also, experimental results are very preliminary and not properly analyzed.  For example the results in Figure 3 are unclear and need to be discussed in detail in the main text.

---

### Official Review · AnonReviewer2 · 2017-11-28
**interesting ideas but lack of experimentation**

**Rating:** 4
**Confidence:** 4

**Review:**

The paper proposes a neural architecture to map video streams to a discrete collection of objects, without human annotations, using an unsupervised pixel reconstruction loss. The paper uses such object representation to inform state representation for reinforcement learning. Each object is described by a position, appearance feature and confidence of existence (presence). The proposed network predicts a 2D mask image, where local maxima  correspond to object locations, and values of the maxima correspond to presence values. The paper uses a hard decision on the top-k objects (there can be at most k objects) in the final object list, based on the soft object presence values (I have not understood if these top k are sampled based on the noisy presence values or are thresholded, if the authors could kindly clarify).
 The final presence values though are sampled using Gumbell-softmax.

Objects are matched across consecutive frames using non parametric (not learnable) deterministic matching functions, that takes into account the size and appearance of the objects.

For the unsupervised reconstruction loss, a static background is populated with objects, one at a time, each passing its state and feature through deconvolution layers to generate RGB object content.

Then a policy network is trained with deep Q learning whose architecture takes into account the objects in the scene, in an order agnostic way, and pairwise features are captured between pairs of objects, using similar layers as visual interaction nets.

Pros
The paper presents interesting ideas regarding unsupervised object discovery

Cons:
The paper shows no results. The objects discovered could be discovered with mosaicing (since the background is static) and background subtraction.

---

### Decision · Program_Chairs · 2018-01-29
**ICLR 2018 Conference Acceptance Decision**

**Decision:**

Reject

**Comment:**

All three reviewers recommended rejection and there was no rebuttal.